# Increased burden of ultra-rare structural variants localizing to boundaries of topologically associated domains in schizophrenia

Matthew Halvorsen[1,10], Ruth Huh[2,10], Nikolay Oskolkov [3,10], Jia Wen [1,10], Sergiu Netotea[4], Paola Giusti-Rodriguez [1], Robert Karlsson [5], Julien Bryois[5], Björn Nystedt [6], Adam Ameur [7], Anna K. Kähler[5], NaEshia Ancalade[1], Martilias Farrell [1], James J. Crowley[1,8,9], Yun Li[1,2], Patrik K. E. Magnusson [5], Ulf Gyllensten[7], Christina M. Hultman[5], Patrick F. Sullivan[1,5,8✉] & Jin P. Szatkiewicz [1,8✉]

Despite considerable progress in schizophrenia genetics, most findings have been for large rare structural variants and common variants in well-imputed regions with few genes implicated from exome sequencing. Whole genome sequencing (WGS) can potentially provide a more complete enumeration of etiological genetic variation apart from the exome and regions of high linkage disequilibrium. We analyze high-coverage WGS data from 1162 Swedish schizophrenia cases and 936 ancestry-matched population controls. Our main objective is to evaluate the contribution to schizophrenia etiology from a variety of genetic variants accessible to WGS but not by previous technologies. Our results suggest that ultra-rare structural variants that affect the boundaries of topologically associated domains (TADs) increase risk for schizophrenia. Alterations in TAD boundaries may lead to dysregulation of gene expression. Future mechanistic studies will be needed to determine the precise functional effects of these variants on biology.

[1] Department of Genetics, University of North Carolina, Chapel Hill, NC 27599, USA. [2] Department of Biostatistics, University of North Carolina, Chapel Hill, NC 27599, USA. [3] Department of Biology, National Bioinformatics Infrastructure Sweden, Science for Life Laboratory, Lund University, 22362 Lund, Sweden. [4] Department of Biology and Biological Engineering, National Bioinformatics Infrastructure Sweden, Science for Life Laboratory, Chalmers University of Technology, 41258 Göteborg, Sweden. [5] Department of Medical Epidemiology and Biostatistics, Karolinska Institutet, 17177 Stockholm, Sweden. [6] Department of Cell and Molecular Biology, National Bioinformatics Infrastructure Sweden, Science for Life Laboratory, Uppsala University, 75237 Uppsala, Sweden. [7] Department of Immunology, Genetics and Pathology, Science for Life Laboratory, Uppsala University, 75185 Uppsala, Sweden. [8] Department of Psychiatry, University of North Carolina, Chapel Hill, NC 27599, USA. [9] Department of Clinical Neuroscience, Karolinska Institutet, 17177 Stockholm, Sweden. [10]These authors contributed equally: Matthew Halvorsen, Ruth Huh, Nikolay Oskolkov, Jia Wen. ✉email: patrick.sullivan@ki.se; jin_szatkiewicz@med.unc.edu

Since the first major study over 70 years ago[1], twin, family, and adoption studies have strongly and consistently supported the existence of a genetic basis for schizophrenia[2–4]. Its inheritance is complex with both genetic and non-genetic contributions indicated by estimates of pedigree-heritability (60–65%)[3,4] and twin-heritability (81%)[2] that are well under 100%. Although these genetic epidemiological results were fairly consistent, their validity was dependent on multiple assumptions and contained specific information about genetic architecture.

In the past decade, genome-wide association (GWA) studies that genotyped hundreds of thousands of single-nucleotide polymorphisms (SNPs) in tens of thousands of cases and controls have directly evaluated the common-variant SNP-heritability of schizophrenia[5–7]. In the most recent study of 40,675 schizophrenia cases and 64,643 controls, the SNP-heritability of schizophrenia was 24.4% (SE 0.0091, liability scale), and 145 significant loci were identified[6]. SNP array data can also be used to assess rare copy number variants (CNVs). In the largest study to date of 21,094 cases and 20,227 controls[8], eight CNVs reached genome-wide significance: CNV deletions at 1q21.1, 2p16.3 (NRXN1), 3q29, 15q13.3, and 16p11.2 (distal) and 22q11.2 plus CNV duplications at 7q11.23 and 16p11.2 (proximal). These events were uncommon and any one of these eight CNVs were present in 1.42% of cases and 0.15% of controls. There is evidence that rare coding single-nucleotide variants (SNVs) and insertion–deletions (indels) contribute to risk in a low percentage of cases although few genes have been implicated from exome sequencing[9–11].

Thus, after a decade of increasingly larger studies, the discovered genetic variants that confer risk for schizophrenia are primarily common variants with subtle effects on risk[6,7,9,10]. The interpretation of common variant findings is markedly improved via the addition of functional genomic data from brain[7,12,13]; nonetheless, there remains a gap between the pedigree- and twin-heritability estimates for schizophrenia and its SNP-heritability. Some argue that this gap is irrelevant as these different types of heritability are incompatible and as biological insights have always been the core goal of GWA for schizophrenia rather than accounting for twin/pedigree heritability. It is also possible that the heritability gap is informative, that SNP array and WES are missing etiologically important genetic variation. GWA genotyping directly captures 500K-1M SNPs followed by imputation to indirectly assess 7–10 M variants. This process is imprecise as some regions of the genome are not well covered, and some non-SNP types of genetic variation are missed. WES provides data on the protein-coding fraction of the genome (~3%) and will miss many regulatory features.

By evaluating high-coverage WGS data on 21,620 individuals in the TOPMed study, Wainschtein et al.[14] reported recovery of nearly all of the pedigree heritability for height and body mass index. The missing heritability was found to reside in rarer genetic variation (minor allele frequency (MAF) 0.0001–0.1) in regions of relatively low linkage disequilibrium (LD) and often outside of protein-coding portions of the genome. The fundamental reason for the missing heritability of height and body mass may merely be technical: the least expensive technologies only partly assess the genome with inexpensive SNP arrays capturing common variants in high LD regions and WES capturing much of the known protein-coding genome. The Wainschtein et al. finding is consistent with prior observations that rarer and evolutionarily younger SNPs have higher SNP-heritability for multiple complex traits[15].

To capture genetic variation as comprehensively as possible, WGS is required. WGS provides nucleotide-level resolution throughout the accessible genome along with detection of most structural variants (SVs). Many types of genetic variation are discoverable by WGS without regard to local LD, and these include SNVs and indels in low LD regions, uncommon or rare regulatory variants, rare SVs missed by SNP arrays and WES due to small size or complexity, and common SVs missed by SNP arrays. The NHLBI TOPMed Program recently published high-coverage (30×) WGS data of 53,831 diverse individuals that included ~381 M SNVs and ~29 M indels[16]. TOPMed WGS identified 16% more variants than low-coverage WGS (6×), with essentially all new variants being rare (MAF < 0.005); and 17% more coding variants than both low-coverage WGS and WES (30×). The distribution of variant sites in TOPMed WGS revealed that the vast majority of human genetic variation is rare and noncoding. There are a few published WGS studies of schizophrenia (Supplementary Table 1). Of these studies, many employed family-based designs and the largest case–control WGS study had 321 schizophrenia cases and 148 controls.

In this study, we analyze high-coverage WGS from 1162 schizophrenia cases and 936 ancestry-matched population controls. WGS data are generated using identical protocols at the same facility and all WGS data are jointly processed and analyzed. The schizophrenia cases also have SNP array[17,18] and exome sequencing data[9,10] which is compared to WGS to assess data quality. Our main objective is to evaluate the contribution to schizophrenia etiology from variants that are revealed by WGS but not by GWA and WES. To quantify phenotypic variance explained by rare variants, we estimate heritability using WGS. To identify the role of noncoding variants, we focus on empirically determined maps of sequence constraints[19,20] and functional genomic annotations generated in human brain[12,13]. We particularly focus on ultra-rare variants as this frequency class has a notable impact on schizophrenia risk in WES and CNV studies[8,10]. We replicate key prior reported excess in schizophrenia of loss-of-function (LOF) ultra-rare sequence variants in LOF-intolerant genes. We find an increased burden in schizophrenia of ultra-rare SVs that affect the boundaries of topologically associated domains.

## Results

**Overview**. Figure 1 gives an overview of the study. Our workflow was designed to evaluate the contribution of directly genotyped genetic variation across the allelic spectrum and evaluate genetic variation missed by prior approaches.

**Study samples and sequencing**. Following quality control, we analyzed WGS on 1162 schizophrenia cases and 936 ancestry-matched population controls from Sweden (total 2098 subjects). Cases were selected to have typical Swedish ancestry, unequivocal schizophrenia case status, and without a known pathogenic CNV (e.g., 22q11 deletion). Controls were group matched to cases by ancestry. The median WGS coverage per sample was 36.62 reads per base (Supplementary Fig. 1). For each group, we constructed a curve for mean fraction of bases covered deeper than a specified threshold as a function of depth of coverage. The shapes of the mean curves were similar between cases and controls (Supplementary Fig. 2). Principal components analysis confirmed the relative homogeneity of the sample (Supplementary Fig. 3).

We took multiple steps to minimize chances of spurious associations with schizophrenia: (1) WGS for all subjects was performed at the same facility using identical procedures; (2) all WGS data were jointly processed; (3) variant calling was conducted jointly for all subjects; (4) all subjects were ethnic Swedes of similar empirical ancestry (Supplementary Fig. 3); and (5) in association analyses, we controlled for empirically determined potential confounders to mitigate impact on spurious association signals (Methods). As discussed more fully below, we did not find evidence

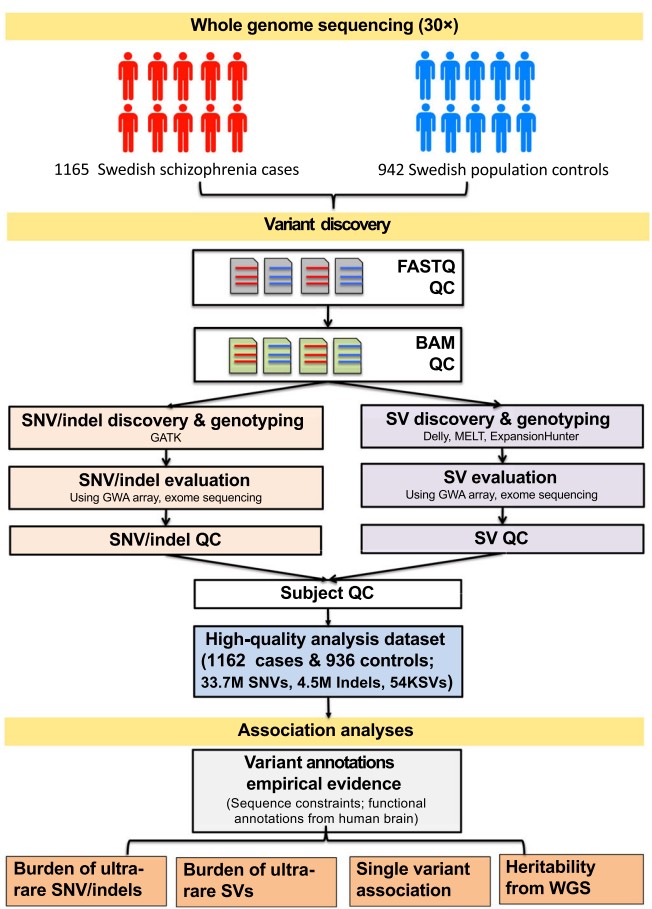

**Whole genome sequencing (30×)**

1165 Swedish schizophrenia cases 942 Swedish population controls

**Variant discovery**

FASTQ QC

BAM QC

SNV/indel discovery & genotyping
GATK

SV discovery & genotyping
Delly, MELT, ExpansionHunter

SNV/indel evaluation
Using GWA array, exome sequencing

SV evaluation
Using GWA array, exome sequencing

SNV/indel QC

SV QC

Subject QC

High-quality analysis dataset
(1162 cases & 936 controls;
33.7M SNVs, 4.5M Indels, 54KSVs)

**Association analyses**

Variant annotations
empirical evidence
(Sequence constraints; functional
annotations from human brain)

Burden of ultra-rare SNV/indels | Burden of ultra-rare SVs | Single variant association | Heritability from WGS

**Fig. 1 Overview of WGS analysis.** WGS data were generated using identical protocols at the same facility and all WGS data were jointly processed and analyzed. The schizophrenia cases also had GWA SNP array and exome-sequencing data for comparison for the purpose of quality assessment. We started with 1165 schizophrenia cases and 942 ancestry-matched population controls. After QC, 1162 cases and 936 controls remained. Variant annotation focused on empirically determined annotation methods.

of inflation (e.g., for common variant case-controls tests, $\lambda_{GC}$ was 1.03 and LD score regression intercept was 0.997 (SE = 0.0065), which are inconsistent with systematic biases).

**WGS variant identification.** SNV/indels: We detected 33,746,530 SNVs and 4,551,507 indels across the autosomes of the 2098 cases and controls. Individual subjects had a mean of 2,358,544 SNVs (range: 2,063,297–2,513,607), and 383,929 indels (range 329,678–409,893; mean insertion size 3.09 bp and mean deletion size 3.59 bp). Of the full set of unique SNVs and indels detected in the WGS data, 45.43% of SNVs and 37.03% of indels were detected in only one individual as heterozygotes (singletons). These data included many variants not found in imputation reference panels. For example, when requiring exact match for chromosome, position, reference, and alternative allele, 15,688,760 SNVs are not in Haplotype Reference Consortium (HRC r1.1) reference panel[21] (stratified by MAF: 286,599 MAF > 0.05, 163,346 MAF 0.005–0.05, 15,238,815 MAF < 0.005); 21,574,998 SNVs are not in 1000 Genomes Project phase 3 (1000GP p3v5) reference panel[22] (61,993 MAF > 0.05, 309,989 MAF 0.005–0.05, 21,203,016 MAF < 0.005), and 12,341,197 SNVs are not in TOPMed[16] Freeze 3a (stratified by MAF: 28,179 MAF > 0.05, 29,233 MAF 0.005–0.05, 12,283,785 MAF < 0.005). We also called 57,785 SNVs and 8270 indels on chrX, and

subjects had a mean of 6084 SNVs (range 5035–7183) and 1753 indels (range 1334–2091).

To evaluate the capacity of WGS to detect SNVs or indels, we compared our WGS data to independent exome sequencing data on 1154 of the 1162 schizophrenia cases[10]. We estimated genotype accuracy by calculating the concordance rate between genotypes from WGS and WES[10] for all autosomes. For SNVs, genotype accuracy was 0.9999, 0.999, and 0.997 for homozygous reference, heterozygous, and homozygous non-reference genotypes (Supplementary Table 2a). For indels, genotype accuracy was 0.998, 0.984, and 0.984 for homozygous reference, heterozygous, and homozygous non-reference genotypes (Table S2a). When stratified by MAF, genotype accuracy estimates were consistent across common, low-frequency, rare, and ultra-rare variants, and similar to the overall genotype accuracy (Supplementary Table 2b–e).

SVs: We detected 17,895 deletion (DEL) sites, 4129 tandem duplication (DUP) sites, 4458 inversion (INV) sites, and 27,808 mobile element insertions sites (MEI, including 23,432 ALU, 1429 SVA, and 2956 LINE1). The sizes of DEL, DUP, and INV ranged from 500 bp to 1 Mb, with median sizes of 2592 bp for DEL, 7179 bp for DUP, and 3265 bp for INV (Supplementary Fig. 4). The sizes of MEI ranged from 15–6019 bp, with a median size of 279 bp for ALU, 1162 bp for SVA, and 1780 bp for LINE1 (Supplementary Fig. 5). For any non-reference genotype, subjects carried a mean of 1241 DEL (range 657–1357), 183 DUP (range 157–209), 373 INV (range 321–878), 2663 ALU (range 2077–3439), 82 SVA (range 56–107), and 249 LINE1 (range 196–302).

To evaluate the capacity of WGS to detect SVs, we compared WGS data to prior copy number variant data from GWA SNP array[18] and WES[23] on 1085 of the 1162 schizophrenia cases. First, INV, MEI, and common SVs are largely inaccessible to SNP arrays[18] and WES studies[23]. Second, prior GWA SNP array studies were limited to deletions and duplications >100 kb; however, >95% of DEL and >77% DUP detected from WGS were <20 kb. Consequently, SNP arrays found only 3.5% of DEL variants and 17.7% of DUP variants found by WGS (requiring 50% reciprocal overlap). Third, when restricted to exons, WES found only 13.7% of exonic DEL and 35.6% of exonic DUP variants found by WGS (based on 50% reciprocal overlap). Finally, for DEL and DUP variants that are comparable between technologies, we computed concordance rates between WGS and SNP array or WES (Supplementary Table 3). When compared to SNP arrays, we estimated that the concordance rate was 0.992 for DEL and 0.965 for DUP. When compared to WES, we estimated that the concordance rate was 0.987 for DEL and 0.967 for DUP.

Repeat expansions: WGS can detect pathogenic disease-associated repeat expansions (e.g., the *HTT* CAG repeat that causes Huntington's disease), which are inaccessible to SNP arrays. We screened our samples for repeat expansions in 16 genes that are established causes of disease, and found that 16 cases and 7 controls had modest repeat expansions just within the predicted pathogenic range (Supplementary Table 4). Because no case or control had a register diagnosis consistent with these generally highly penetrant disorders, we assumed these were false positives or the modest repeat expansions were not long enough to cause disease.

**Burden analysis of ultra-rare SNV/indels.** Consistent with recent studies[10], we focused on ultra-rare sequence variants (URVs) including ultra-rare SNVs and indels. We defined URVs as found once in the WGS case/control cohort and absent from independent population cohorts (i.e., gnomAD r2.0.2 allele count = 0 and non-psychiatric subset of ExAC r0.3 allele count = 0)[24,25]. From theory[26] and our calculations (Supplementary Fig. 6), power is low for single-variant analysis for MAF < 0.01. Collapsing methods are

key approaches for rare variants and can enhance power by accumulating information across different rare variants that impact a gene/locus or a set of genes/loci[27]. We used burden testing as the primary analytical tool to contrast cases and controls for total event counts in genomic loci of interest. Burden testing is appropriate when most variants across a set of genetic loci impact phenotype in the same direction and with similar magnitude[27]. We estimated statistical power for burden tests and found that we had ≥80% power to detect association of URVs when the aggregated minor allele count (MAC) was 20 (i.e., aggregated MAF = 0.01) and the genotypic relative risk was ≥4.9 (assuming a type I error level of $1 \times 10^{-5}$). As a final step in quality control and following an approach previously established in the full Swedish sample[10], we pruned samples that had an outlier total URV count mostly because of relatively higher ancestry heterogeneity[10] (Methods, Supplementary Figs. 7 and 8). We conducted burden analyses of URVs in 1104 cases and 921 controls (mean URV counts in cases vs controls: 4262 vs 4249, P = 0.4225, Supplementary Fig. 7). The total number of qualifying URVs in these samples was 8,073,782, of which 7,991,557 (98.9%) were noncoding. Full results are listed in Supplementary Table 5 and summarized below. For multiple-testing adjustment, we applied the Benjamini and Hochberg false discovery rate (BH-FDR) method to the family of hypotheses involving ultra-rare SNV/indels which included a total of 74 tests (Supplementary Table 5).

Confirmation of prior results: We first evaluated the prior WES finding that schizophrenia cases have an excess of damaging protein-coding URV (odds ratio [OR] = 1.07; 4877 cases and 6203 controls)[10]. As shown in Fig. 2, we found an excess of LOF URVs in schizophrenia cases (OR = 1.082, P = 0.0002, BH-FDR multiple-testing adjusted P = 0.0049). This excess was notable (OR = 1.203, P = 0.0005, adjusted P = 0.0092) in genes that are intolerant to LOF variation (defined as pLI > 0.9 in the non-psychiatric subset of ExAC[24], where pLI is the probability that a gene is intolerant to a LOF mutation). Increased burden was

prominent in the subset of LOF-intolerant genes that are risk genes from WES for neurodevelopmental disorders[11] (OR = 2.983, P = 0.0011, adjusted P = 0.0163). A key advantage of WGS over WES for protein-coding regions is independence of design, coverage, and performance of exome capture baits[16]. The exome capture baits used in WES are imperfect, however, after multiple testing correction, we did not find any significantly increased burden of coding URVs outside of targeted exonic sequences of LOF-intolerant genes (Supplementary Fig. 9, Supplementary Table 5).

Burden analysis of noncoding ultra-rare SNV/indels in constrained regions: We defined variants as putatively noncoding if they did not alter sequence content of coding regions or splice dinucleotides of GENCODE protein-coding transcripts. These noncoding variants may confer risk via a variety of mechanisms (e.g., by altering an unannotated protein-coding transcript, untranslated regions, splicing, transcription factor binding, or an epigenetic site). We evaluated burden of noncoding variants that are more likely to be deleterious by focusing on ultra-rare noncoding variants that are likely to be subject to purifying selection in a manner similar to coding URVs. We compared case/control burden of noncoding URVs across binned regions by sequence constraint for the human species[19] and by constraint across mammalian species[20] (Supplementary Fig. 10). The human constraint was built upon the context-dependent tolerance score (CDTS) which indicates the degree of depletion of genetic variation at the population level using 11,257 human genomes (the lower the percentile rank of CDTS the more constrained the region)[19]. The mammalian constraint was based on genomic evolutionary rate profiling (GERP) score which quantifies substitution deficits in multiple alignments (the higher the GERP score, the more constrained the region)[20]. We concentrated subsequent noncoding URV analyses on variants in regions that were highly constrained according to one of these two metrics (CDTS < 1% or GERP ≥ 4) due to prior observation that the overlap between CDTS (conservation in the current human population) and GERP (interspecies conservation) was limited and heavily enriched for protein-coding regions[19]. We did not observe a case excess of noncoding URVs that survived multiple test correction based on this criterion alone (OR = 1.009, P = 0.0342, adjusted P = 0.2819, Supplementary Fig. 10).

Burden in annotations experimentally derived from human brain: Annotations from appropriate tissues help predict functional variants[7,28]. We compared case/control burden of noncoding URVs in constrained regions (as defined above CDTS <1% or GERP ≥ 4) within functional annotations experimentally derived from human brain tissue known to effect gene expression. These annotations include open chromatin regions from ATAC-seq, frequently interacting regions (FIREs), topologically associating domains (TADs), and chromatin interactions from Hi−C; epigenetic marks from ChIP-seq (CTCF, H3K27ac, and H3K4me3). We also included annotations of brain-expressed exons identified from long-read RNA-seq data[29], as constrained noncoding URVs inside brain exons could impact functional noncoding elements within untranslated regions of annotated transcripts or protein-coding sequences from unannotated transcripts. We did not identify any single annotation with a significant case excess of URVs within constrained regions (Supplementary Fig. 11, Supplementary Table 5).

Burden in promoter regions: A recent study focused on de novo SNV/indels found evidence for a contribution to autism spectrum disorder from variants in constrained nucleotides within promoter regions[30]. Defining promoter regions the same way as in An et al.[30] (2 kilobases (kb) upstream of an annotated transcription start site), we compared case/control burden of noncoding URVs within constrained nucleotides (as defined above) in promoter regions of genes that are putatively

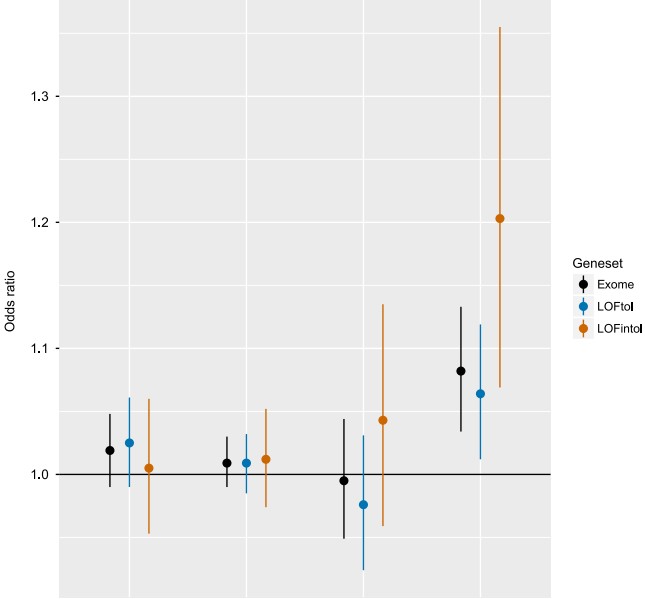

**Fig. 2 Burden of coding ultra-rare SNVs and indels.** X-axis: annotation class. Y-axis: odds ratio. Legend: exomes: coding variants in all genes; LOFtol: in genes tolerant to loss-of-function variation; LOFintol: in genes intolerant to loss-of-function variation. For each specific burden test, we used a vertical line to indicate the 95% confidence interval of odds ratio and a dot at the center of the vertical line to indicate the point estimate of odds ratio.

LOF-intolerant (as defined above). No significant case excess was observed (OR = 0.966, $P$ = 0.9812, adjusted $P$ = 0.9907). A similar result was obtained when performing this test specifically on the subset of LOF-intolerant genes previously described as neurodevelopmental risk genes[11] (OR = 0.99, $P$ = 0.5551, adjusted $P$ = 0.6956). To take the three-dimensional genome into account, we used brain chromatin interaction data to identify any cis-regulatory elements (e.g. promoter and enhancers) connected with LOF-intolerant genes. No significant case excess was observed (OR = 1.006, $P$ = 0.1904, adjusted $P$ = 0.427).

X chromosome: We tested male cases and controls to determine if the coding variant excess replicated in chrX genes. We did not detect a significant difference in synonymous variant burden or LOF variant burden but note that power was low.

**Burden analysis of ultra-rare SVs**. We performed analyses of ultra-rare SVs on the full sample (1162 cases and 936 controls) and on the subsample used for URV burden testing (1104 cases and 921 controls). Note that cases with known pathogenic CNVs or unusually high CNV burden were excluded[18]. Because all results were similar, we report the analysis results using the full sample. The total number of ultra-rare SVs in our sample was 6809 for DEL, 1917 for DUP, and 729 for INV. The sizes of these ultra-rare SVs were smaller than those from SNP arrays (DEL mean 15.2 kb for cases and 13.8 kb for controls; DUP mean 56.5 kb for cases and 52.6 kb for controls; and INV mean 100 kb for cases and 76 kb for controls).

Confirmation of prior results: Higher genome-wide burden of rare SVs in schizophrenia cases has been repeatedly observed in studies using SNP arrays[8,18] (i.e., rare, large SVs with MAF < 0.01 and size > 100 kb). Burden was greater for SVs that were deletions, larger, or rarer. To calibrate our analyses, we verified this general pattern of findings using WGS SV calls (Supplementary Table 6).

Genome-wide burden of ultra-rare SVs: Using the DEL, DUP, and INV genotypes described above, we evaluated the genome-wide burden of ultra-rare SVs (Supplementary Fig. 12 and Supplementary Table 7). We defined ultra-rare SVs as found once in the WGS case/control cohort and absent from independent population cohorts[31,32]. Consistent with previous reports[8,18], ultra-rare DEL were significantly enriched in cases (OR = 1.086, $P$ = 0.0001, BH-FDR multiple-testing adjusted $P$ = 0.0029). The burden of ultra-rare DUP and INV were similar between cases and controls (DUP: OR = 1.06, $P$ = 0.0920, adjusted $P$ = 0.2052; INV: OR = 1.015, $P$ = 0.2903, adjusted $P$ = 0.4009). Most of these ultra-rare SVs were noncoding (Supplementary Fig. 13, 87.2% for DEL, 71.1% for DUP, and 89.4% for INV). When stratified by coding/noncoding status, the results were similar (Supplementary Table 7).

Burden in epigenomic annotations from human brain: We hypothesized that the elevated genome-wide burden of ultra-rare SVs may be partitioned across functional elements with evidence for gene regulation in the brain[13]. We focused on ultra-rare SVs that intersected ≥10% of the functional elements (Fig. 3, Supplementary Table 8). Burden tests found a significant enrichment of ultra-rare SVs in schizophrenia cases that impacted TAD boundaries from adult (OR = 1.613, $P$ = 0.0037, adjusted $P$ = 0.0283) and fetal brain (OR = 1.581, $P$ = 0.0039, adjusted $P$ = 0.0283). No significant enrichment was found for any other class of functional elements. TAD boundaries have been shown to be under purifying selection. Multiple studies suggest that altering TAD boundaries results in the disarrangement of enhancer and promoter contacts, thus impacting local gene expression. Disruption of TAD boundaries by SVs have been associated with developmental disorders[33,34].

Burden in regulatory elements connected with schizophrenia risk loci: We hypothesized that the elevated genome-wide burden of ultra-rare SVs may be partitioning to regulatory elements within schizophrenia risk loci. To take the three-dimensional genome into account, we used chromatin interaction data from adult brain to identify regulatory elements connected with schizophrenia risk loci, capturing any empirically defined cis-elements either nearby or distal[13]. As above, we performed a burden test using the 10% overlap criterion for any ultra-rare DEL, DUP, or INV in the regulatory elements of these schizophrenia risk loci. No significant enrichment in schizophrenia cases was found (Fig. 3, Supplementary Table 8).

**Validation and analysis of ultra-rare TADs-affecting SVs**. To gain a deeper understanding, we followed up on the finding of significantly increased burden of ultra-rare SVs that affected TAD boundaries. We found that a higher rate of variants in cases versus controls was present when those variants were stratified by coding or non-coding status (Supplementary Fig. 14, Supplementary Table 9) or by variant type (i.e., DEL, DUP, or INV; Supplementary Fig. 15, Supplementary Table 10). Burden was greater for those variants that were DEL, or had larger overlap with TAD boundaries.

Next, we attempted to verify the validity of those TADs-affecting ultra-rare DEL and DUP that were detected in schizophrenia cases. First, we looked up the GWA array data in the same samples (Supplementary Table 11). We found that 27.9% of these DEL and 52.6% of these DUP were concordant with GWA array data (50% reciprocal overlap) and were additionally confirmed by inspecting their WGS read alignments using IGV[35] (Supplementary Figs. 16 and 17). The remaining variants that were not found from GWA array data were notably smaller in size (median 7.6 kb) than those concordant (median 181 kb), suggesting that they may have been missed by GWA array technology. Second, for variants not verifiable using GWA arrays, we manually inspected their WGS read alignments using IGV[35] (Supplementary Figs. 18 and 19), and all were confirmed.

Finally, we evaluated genomic features nearby those TADs-affecting ultra-rare SVs that were detected in schizophrenia cases (Supplementary Table 12). We found that these SVs span 4 – 995 kb and 71% of them (67 out of 94) overlapped ≥1 genes. There was a notable difference between TADs-affecting ultra-rare DEL and DUP: 44.7% (17 out of 38) of DUP overlapped genes had high pLI scores or were genes implicated in schizophrenia or neurodevelopmental disorders, whereas 16.3% (7 out of 43) of DEL overlapped genes had high pLI scores or were implicated in neurodevelopmental disorders ($H_0$: no difference between DEL and DUP, Fisher's exact test $P$ = 0.0072). Furthermore, 36.8% (14 out of 38) of the DUP connected with 43 genes with high pLI scores or implicated in neurodevelopmental disorders via a high-confidence regulatory chromatin interaction (HCRCI); whereas 18.6% (8 out of 43) of the DEL connected with 18 genes with high pLI scores or implicated in neurodevelopmental disorders via a HCRCI (Fisher's exact test $P$ = 0.0824). Our observations are consistent with a previous report that duplications display a more complex relationship with chromatin features than deletions[36]. INV was similar to DUP (Supplementary Table 12; $H_0$: no difference between INV and DUP, Fisher's exact test $P$ = 0.53).

**Common variants with large effects were not identified**. Because SNP arrays do not cover the entire genome even with imputation, we performed single-variant association analysis for all common variants obtained from WGS. Given the sample size of 2098 (1162 cases and 936 controls), we estimated that our sample had ≥80% power to detect risk variants with MAF = 0.25 and genetic relative risks ≥2.0, assuming a type I error level of $5 \times 10^{-8}$ (Supplementary Fig. 6).

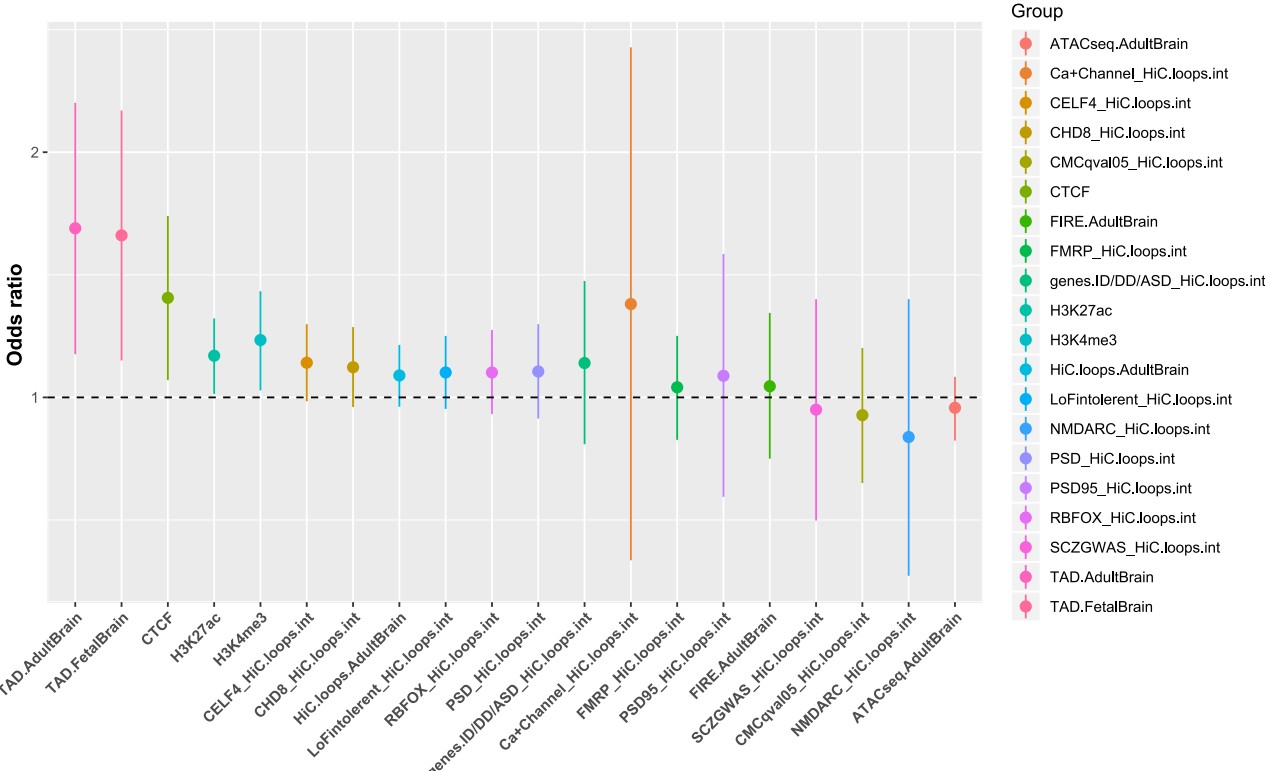

**Fig. 3 Burden of ultra-rare SVs in brain epigenomic annotations and related analysis.** For each specific burden test, we use a vertical line to indicate the 95% confidence interval of odds ratio and a dot at the center of the vertical line to indicate the point estimate of odds ratio. Labels on X-axis indicate the specific annotations that were considered. "TADbou": TAD boundaries. Epigenomic annotations include TADbou.AdultBrain, TADbou.FetalBrain, ATACseq.AdultBrain, FIRE.AdultBrain, CTCF, H3K27ac, and H3K4me3. Regulatory elements connected with schizophrenia risk loci are labeled as "gene.set.name_HiC.loops.int". For example, "CELF4_ HiC.loops.int" means regulatory elements of the CELF4 gene set identified via chromatin interaction (a.k.a HiC loops) data in human brain. Detailed information about gene sets considered can be found in Methods. Amongst the loci tested, only TAD boundaries derived from both fetal and adult brain tissue showed a significant degree of evidence for excess in cases relative to controls.

SNV/indels: We analyzed 7,895,148 SNVs and 1,368,675 indels with MAF > 0.01 for association with schizophrenia (Supplementary Fig. 20). We obtained a $\lambda_{GC}$ value of 1.03 and LD score regression intercept of 0.997 (SE = 0.0065), indicating no departure from null expectations or uncontrolled bias. Single-variant association analysis was done using logistic regression assuming an additive genetic model including PC2 as covariate for autosomes, and sex and PC2 as covariate for chrX. A number of variants exceeded genome-wide significance but, upon review, all 15 were false positives due to lack of read alignment support. These results are consistent with accumulated experience in schizophrenia genomics where larger sample sizes are required to detect common variant associations[7]. We believe that this null result is important: we have excluded the possibility of common variants (MAF > 0.01) with large effects in less accessible parts of the genome that have not been evaluated by GWA SNP arrays.

SVs: Association analysis of common SVs has the potential to identify causative mutations leading to actionable findings, and much of this class of variants is inaccessible to SNP-based studies. Here we performed association analysis for SVs with MAF > 0.01, using logistic regression models and covariates as described above. The main analysis was for 2199 common DEL (Supplementary Fig. 21) but no association reached genome-wide significance. We then inquired into common DUP, INV, ALU, LINE1, and SVA, but also found no significant associations (Supplementary Figs. 22–26).

**Heritability estimation using WGS.** Heritability is the proportion of phenotypic variance explained by genetic factors.

Understanding the sources of missing heritability for schizophrenia – the discrepancy between pedigree-heritability of 60–65%[3,4] and common-variant SNP-heritability of 24%[6] – is important for experimental designs to identify additional trait loci and possibly for subsequent precision medicine initiatives. Using WGS data for height and body mass index, Wainschtein et al. recently found WGS-heritability very close to twin/pedigree heritability[14]. WGS allowed them to include effects in genomic regions of low MAF and low LD, precisely the regions that are poorly captured by typical SNP arrays or imputation.

Following Wainschtein et al.'s approach[14], we estimated schizophrenia heritability from our WGS data using 1151 cases and 911 controls (post-QC subjects and pairwise genetic relatedness < 0.05), and 17,364,971 sequence variants (post-QC autosomal SNV/indels observed ≥ 3 times or MAF ≥ 0.0007). To evaluate the effect of progressive inclusion of more variants, we computed heritability in different ways by selecting WGS variants that corresponds to variant locations in HapMap3[37], those imputable from 1000 G p3v5[22] and HRC r1.1[21], and finally by including all WGS variants.

First, we assessed common SNP-heritability in the WGS sample using the GREML single-component method implemented in GCTA[38,39]. Using 1,189,077 SNPs from WGS that corresponds to the SNP locations in HapMap3, the SNP-heritability was 0.45 (standard error [SE] 0.089, liability scale assuming lifetime risk of 1%). Using 7,141,717 SNV/indels from WGS that corresponds to the variant locations imputable from 1000GP p3v5, the SNP-heritability was 0.48 (SE 0.091). These estimates are numerically greater than that estimated

from SNP arrays in the full Swedish sample (5001 cases; GCTA SNP-heritability using HapMap3 data: 0.32, SE 0.03, and using 1000 Genomes data: 0.33, SE 0.03)[6], presumably due to the fact that more stringent evidence of schizophrenia was used for samples selected for WGS than that in the full sample (Methods).

Next, we evaluated SNP-heritability using 8,498,854 SNV/indels from WGS that corresponds to the variant locations imputable from HRC r1.1. We used the recommended GREML-LDMS method in GCTA[39,40] because it is unbiased regardless the properties (e.g. MAF and LD) of the underlying causal variants (Supplementary Fig. 27a). The estimated SNP-heritability was 0.52 (SE 0.22).

Finally, we used all sequence variants (17,364,971 as above) from WGS and the GREML-LDMS method[39,40] to estimate WGS-heritability and partition additive genetic variance. We found the estimated WGS-heritability was 0.56 (SE 0.51). The point estimate of 0.56 is closer to pedigree-heritability (0.6–0.65, refs. [3,4]), but the SE is large. For rare variants with MAF 0.0007–0.01, WGS variants in the low-LD group contributed to 0.40 of the phenotypic variance whereas variants in the high-LD group contributed to 0.01 of the variance (Supplementary Fig. 27b). In contrast, for HRC-imputable variants, 0.06 and 0.03 of the phenotypic variance was contributed by variants in the low- and high-LD groups for MAF 0.0007–0.01 (Supplementary Fig. 27a). The contribution to phenotypic variance from rare variants in low-LD with nearby variants was only revealed by WGS. These variants could only be directly assayed by WGS as they are not present in SNP arrays and their imputation is not accurate[14].

In sum, the point estimates for heritability were progressively larger as we included more variants and there was a sizable contribution from rare variants with low-LD metrics that are accessible only via WGS. However, our estimates of SNP- and WGS-heritability had large standard errors. This was due to limited sample size and case-control study design (i.e. not continuous trait as height or body mass). The WGS-heritability estimate had the largest SE which was additionally due to the large number of rare variants with low MAF and low LD. The sampling variance of SNP-based heritability estimate is approximately inversely proportional to sample size and is proportional to the effective number of independent variants[41,42] Furthermore, we likely underestimated WGS-heritability, especially the contribution from rare variants with MAF < 0.001: First, Wainschtein et al.[14] was able to include WGS variants with MAF as low as 0.0001 (corresponding to MAC ≥ 3 in TOPMed data with 21K subjects), whereas in this study we were limited to a minimum MAF of 0.0007 (corresponding to MAC ≥ 3 in 2K subjects). Second, based on a simulation using AbCD[43] assuming 2062 EUR individuals and 30x WGS, we have >99% power to detect variants at MAF > 0.001 but only >53% power to detect variants for MAF 0–0.001. The lowest MAF bin (MAF 0.0007–0.001) that we were able to consider in this study likely included only half of the variants that could have been observed in a sample with 10,000 subjects.

We believe it notable that, although not conclusive, our results for schizophrenia are consistent with those of Wainschtein et al. for height and body mass[14,16]. These results imply that, with larger schizophrenia samples (e.g. a sample size of >33,000 is needed to obtain an SE of 0.02, refs. [14,41,42]), WGS data may be able to fully recover the total additive genetic variance with desired precision and will allow further partitioning of the genome to finer MAF/LD groups as well as a variety of functional annotations[14,42]. The still missing heritability of schizophrenia may be only misplaced, in precisely the blind spots of SNP arrays as has been anticipated for over a decade[44].

## Discussion

We have generated and analyzed a collection of WGS data for a set of patients ascertained for schizophrenia that to our knowledge is the largest described in a publication. The high depth and uniformity of coverage across the genome for these case data allowed us to detect the large majority of genetic variation that are present in the genome, including SNVs, indels, CNVs, mobile element insertions, and inversions. In addition, the availability of similar WGS data from Swedish controls allowed us to systematically measure the burden of these different classes of variation in a case/control manner.

Through the analysis of these data, we were able to replicate key prior reported excess in schizophrenia of LOF URVs in genes that are putatively LOF-intolerant as well as excess of rare deletions genome-wide. This means that we can be more confident that the load of such variants, while modest compared to the identified contribution of common variation to schizophrenia risk, are a subset of the total schizophrenia genetic risk architecture.

Our finding that ultra-rare SVs in schizophrenia cases are enriched at TAD boundaries is not surprising. These variants seem to confer a level of relative risk comparable to protein-coding variants for which we have replicated an excess in schizophrenia. These regions have been reported as being depleted of deletions in human populations relative to the rest of the non-coding genome[36], and clear phenotypic consequences associated with deletion of these elements have already been demonstrated in a number of other diseases[34]. TAD boundaries are critical to the formation and maintenance of chromatin structure[13]. The disruption of these boundaries has the potential to rearrange spatial orientation of regulatory elements that are needed for proper expression, as well as lead to the formation of entirely new TADs. Functional examples of such effects have already been described in mouse models for limb malformation[33]. Based on these prior observations it is unsurprising that of all noncoding loci, the burden of these SVs appears to be highest relative to controls in TAD boundaries.

While our data support an excess of TAD-affecting ultra-rare SVs in schizophrenia cases relative to controls, the precise impact of these variants on gene expression and regulation has yet to be determined. Many of these SVs overlapped genes including some of the risk genes for neuropsychiatric disorders. Mechanistic studies are needed to clarify the precise genomic consequences of these TADs-affecting SVs in human brain. A possible future investigation would be to work with patient derived cells with these TADs-affecting SVs that we have identified and figure out what promoter-enhancer pairing looks like, and if there are any potential changes in gene expression. Our study has highlighted a specific hypothesis for future functional analyses. It will be critical to determine the precise functional effects of these variants on biology, which, in a manner similar to common variant risk, are likely to converge on higher order architectures of gene regulation[7].

We chose not to analyze rare mobile element insertions because variant calling for these variants appear to be noisy from our 30× WGS and there was a lack of external dataset or analytic approach for the need of quality control. Increased somatic L1 insertions have been recently reported in neurons of schizophrenia patients using postmortem brain tissues[45]. The detection of somatic L1 insertions required very deep WGS (e.g. 200×) and tailored analytic methods (e.g. machine learning[45]). For similar reasons, we also chose not to evaluate translocations and complex SVs in this study as we feel that these variants can be better detected from WGS using long-insert jumping libraries, deeper coverage, and targeted capture of breakpoints[46].

The analysis of noncoding variants from WGS data is challenging due to the sheer volume of the noncoding genome and limited methods to predict functional changes[28,30,46]. Recently the

category-wise association study (CWAS) framework has been developed and applied to WGS studies of autism spectrum disorder using 7608 samples from 1902 families[28,30,46]. The CWAS approach applies multiple annotation methods to define tens of thousands of annotation categories each of which are tested for association and accounted for multiple testing. However, there is a trade-off between false positives and false negatives. In this study we adopted the spirit of the CWAS approach and focused on empirically determined annotation methods including (1) conservations of DNA sequence that were estimated from cataloging and comparing genetic variation across human and mammalian species[19,20], (2) multiple epigenomic annotations that were experimentally generated from human brain[12,13], (3) genes and regions that were empirically associated with psychiatric disorders. This approach combined with the relative homogeneity of the Swedish sample helped improve the power to identify functional variants while controlling for false discovery rate.

We failed to detect an excess of risk variation beyond a couple of specific classes of variation, and we believe that this is largely due to a lack of power. Prior data has demonstrated that power to implicate common variation with schizophrenia risk is only sufficient with a much larger case/control cohort, on the order of $N_{case/control} > 10,000$[3,6]. This also applies to implication of genomic loci based on ultra-rare variation. Cohorts larger than ours have failed to implicate burden of ultra-rare coding variants in individual genes with schizophrenia risk[10], and implication of SVs with schizophrenia at locus level resolution required cohorts far larger than ours[8]. Since we can assume that noncoding ultra-rare SNVs and indels will have a smaller relative risk conferred than damaging coding variants, it is clear that implication of this class of variation both across the genome and at locus level resolution will also require a far larger cohort size. Furthermore, larger samples will be necessary to ensure findings are replicable[30].

In sum, to effectively identify the subset of rare variation across the genome that confers schizophrenia risk in patients, we will need to follow the blueprint constructed for common variant GWAS. Substantial collaborative effort will be critical. WGS is expensive and generates a large quantity of sequence data that are difficult to efficiently store and analyze en masse. The financial and computational burden inherent to a case/control WGS analysis with sufficient power for discovery is too much for individual groups or institutions, and will only be feasible through collaborative work in meta-analyzing case/control WGS datasets. The WGS data we have generated are meant to be included in these future efforts.

## Methods

**Ethics**. We have complied with all relevant ethical regulations. The study protocol and all procedures on data from human research subjects were approved by the appropriate ethical committees in Sweden and the US (University of North Carolina [Institutional Review Boards], Karolinska Institutet [Regionala Etikprövningsnämnden, Stockholm], University of Uppsala [Regionala Etikprövningsnämnden, Uppsala]). All participants gave their written informed consent. All genomic coordinates are given in NCBI Build 37/UCSC hg19.

**Subjects**. All schizophrenia cases included this study are from the Swedish Schizophrenia Study (S3). Detailed descriptions of S3 procedures are available elsewhere[17] and are briefly summarized here. S3 cases were identified via the Swedish Hospital Discharge Register that captures >99% of all inpatient hospitalizations in Sweden[47].The register is complete from 1987 and augmented by psychiatric data from 1973 to 1986. The sampling frame is thus population-based and covers all hospital-treated patients. The Hospital Discharge Register contains dates and ICD discharge diagnoses for each hospitalization, and captures the clinical diagnosis made by attending physicians. Case inclusion criteria: ≥2 hospitalizations with a discharge diagnosis of schizophrenia or schizoaffective disorder, both parents born in Scandinavia, and age ≥18 years. Case exclusion criteria: hospital register diagnosis of any medical or psychiatric disorder mitigating a confident diagnosis of schizophrenia as determined by expert review, and included removal of 3.4% of eligible cases due to the primacy of another psychiatric disorder (0.9%) or a general medical condition (0.3%) or uncertainties in the Hospital Discharge Register (e.g., contiguous admissions with brief total duration, 2.2%). The validity of this case definition of

schizophrenia is strongly supported as described in[17]. Ethical committees in Sweden and in the US approved all procedures and all subjects provided written informed consent (or legal guardian consent and subject assent). We also obtained permissions from the area health board to which potential subjects were registered. Potential cases were contacted directly via an introductory letter followed by a telephone call. If they agreed, a research nurse met them at a psychiatric treatment facility or in their home, obtained written informed consent, obtained a blood sample, and conducted a brief interview about other medical conditions in a lifetime.

The S3 included more than 5000 schizophrenia cases, from which we selected 1165 cases for whole-genome sequencing (WGS) in the current study. Our main goal in selection was typical Swedish ancestry and clear schizophrenia caseness. Cases carrying known pathogenic copy number variants (CNVs) (e.g. 22q11del, 16p11dup) were not selected as a primary question of this study is to evaluate the contribution of novel loci on schizophrenia risk. DNA was extracted from peripheral blood samples. Specifically, our selection procedures required the following case inclusion criteria to be met: (1) have high-quality/sufficient DNA that satisfied all criteria: concentration ≥ 80 µg/ml, volume ≥ 150 µl, and purity ratio 1.7–2.2; (2) used in GWA study[17]; (3) have typical Swedish ancestry defined by the first two PCs used in[17]; (4) do not carry known large pathogenic CNVs and are not outliers for total number of CNVs as identified in Szatkiewicz et al.[18]; (5) have stringent evidence of schizophrenia that satisfied all criteria: >8 inpatient or outpatient psychiatric treatment contacts for schizophrenia or schizoaffective disorder, ≥30 inpatient days for schizophrenia, ≥5 redeemed prescriptions for antipsychotics, and few or no treatment contacts for bipolar disorder. Institutional Review Boards at University of North Carolina and regional ethics committee at Karolinska Insitutet (Regionala Etikprövningsnämnden, Stockholm) approved all study procedures and all subjects provided written informed consent.

All control subjects included this study are from the SweGen project, a population-based high-quality genetic variant dataset for the Swedish population. One of the aims of SweGen is to enable WGS association studies for national patient cohorts studies in Sweden, by providing data on well-matched national controls selected on the basis of the genetic structure of the Swedish population. Detailed description of the SweGen subjects are available elsewhere[48] and are briefly summarized here. SweGen project included a total 1000 individuals, of which 942 individuals were selected from The Swedish Twin Registry (STR)[49] and 58 from The Northern Swedish Population Health Study (NSPHS)[50]. Both STR and NSPHS are population-based collections and were approved by local ethics committees. STR is a national registry of Swedish born twins established in the 1960s and, at present, holds information on 85,000 twin pairs. In total, 11,000 individuals from the STR (one per monozygous twin pairs) participated in TwinGene and had existing SNP array genotyping. The Twingene study is a nation-wide and population-based study of Swedish born twins agreeing to participate. The TwinGene sample collection represents the Swedish geographic population density distribution. Based on principal component analysis (PCA), 942 unrelated individuals were selected from TwinGene participants for whole-genome sequencing, mirroring the density distribution. All participants gave their written informed consent and the TwinGene study was approved by the regional ethics committee (Regionala Etikprövningsnämnden, Stockholm, dnr 2007-644-31, dnr 2014/521-32). NSPHS is a health survey in the northern Swedish country of Norrbotten. Based on PCA, 58 individuals were selected from NSPHS. The NSPHS study was approved by the local ethics committee at the University of Uppsala (Regionala Etikprövningsnämnden, Uppsala, 2005:325 and 2016-03-09). All participants gave their written informed consent to the study including the examination of environmental and genetic causes of disease in compliance with the Declaration of Helsinki. Given the selected 1000 subjects that constitutes SweGen, a PCA using genotypes from high-density SNP arrays was performed and confirmed that the SweGen control cohort captured the diversity in the country. Furthermore, since STR and NSPHS are already established national sample collections that do not reflect recent migration patterns, the SweGen control cohort is likely to reflect the genetic structure of Swedish individuals that have been present in Sweden for at least one generation.

From the SweGen subjects, we selected the 942 STR/TwinGene individuals as controls in this study because of their matched ancestry with selected schizophrenia cases. Phenotype data was not allowed in the SweGen project in order to make a less restrictive access policy possible. Consequently, we were unable to screen for the presence of individuals with schizophrenia. However, we estimate that at most 1 control individual may carry a schizophrenia diagnosis (given the estimated schizophrenia prevalence of 0.0009 in the full STR/TwinGene project of 11,000 individuals). Misclassification of a single control subject will not likely affect the results or the power of the study. DNA for the STR/TwinGene individuals was extracted from blood.

All S3 subjects, including those in this WGS study, had GWA SNP array genotyping[17] and exome sequencing[9,10]. DNA was extracted from peripheral venous blood for all subjects. GWAS array genotyping was done in six batches at the Broad Institute of MIT and Harvard using Affymetrix 5.0 (3.9%), Affymetrix 6.0 (38.6%), and Illumina OmniExpress (57.4%). Exome sequencing was done at the Broad Institute of MIT and Harvard in twelve separate waves. The first wave used Agilent SureSelect Human All Exon Kit and Illumina GAII. Other waves used a newer version Agilent SureSelect Human All Exon v.2 Kit and Illumina HiSeq 2000 and HiSeq 2500 instruments. Paired-end reads of 76 bp were used across all waves. Analyses of SNP array and exome sequencing data are previously published.

Data on common SNPs is published in Ripke et al.[17]. Data on exonic SNVs and indels is published in Genovese et al.[10]. Data on large rare CNVs are published in Szatkiewicz et al.[18]. All data are in NCBI build 37/UCSC hg19 coordinates.

**Whole-genome sequencing and data processing**. Library preparation and sequencing was performed by the National Genomics Infrastructure platform in Sweden. All cases and controls were processed using identical library preparation and sequencing protocols at two facilities. WGS libraries were prepared from ~1 µg DNA using Illumina TruSeq PCR-free DNA sample preparation kits targeting an insert size of 350 bp. Library preparation was performed according to the manufacturer's instructions. The protocols were automated using an Agilent NGS workstation and Beckman Coulter Biomek FXp. WGS clustering was done using cBot, and paired-end sequencing with 150 bp read length was performed on Illumina HiSeqX (HiSeq Control Software 3.3.39/RTA 2.7.1) with v2.5 sequencing chemistry.

**Identical analysis pipelines (including software tool versions) were used for processing all case and control samples together**. For alignment, the workflow engine Piper[51] (v1.4.0) was used to perform pre-processing and variant discovery, coordinated using the National Genomics Infrastructure pipeline framework. Following the GATK guidelines, raw reads were aligned to the GRCh37 human reference genome (human_g1k_v37.fasta) using bwa mem[52] (v0.7.12). The resulting alignments (.BAM) were sorted and indexed using SAMtools[53] (v0.1.19). Alignment quality control statistics were gathered using qualimap[54] (v2.2). Alignments for the same sample from different flowcells and lanes were merged using Picard MergeSamFiles (v1.120, https://broadinstitute.github.io/picard).

For quality control of aligned sequence reads, we ran FastQC[55] on the BAM-files in order to understand sequencing quality and to identify outlier samples which might be subject to contamination. We analyzed a number of sequencing QC metrics (e.g., adapter content, per base N nucleotide content, per base sequence content, per base sequence quality, per sequence GC content, per sequence quality scores, sequence duplication level, and sequence length distribution). We analyzed a number of sequence coverage QC metrics produced by SAMtools flagstat (e.g., sequencing depth, percentage of mapped reads, percentage of properly paired reads, percentage of singletons, percentage of duplicates, and percentage of paired end reads with one mate mapped to a different chromosome). Finally, we checked uniformity of read coverage using BEDTools genomecov[56], based on which we required that samples with good coverage have ≥80% of bases be covered at least 20× for confident variant calling. These procedures identified one outlier sample (a schizophrenia case).

We confirmed the identity of all subjects by comparing SNP genotypes from WGS to those from GWA SNP array genotyping[17] and exome sequencing[10]. Identity-by-decent was estimated using PLINK[57] (v1.9) for each sample between WGS-based genotypes and array- or WES-genotypes in overlapping SNPs. Based on this analysis, identity was confirmed for all samples (i.e. no sample swap was found). The identity of SweGen subjects have been confirmed previously in[48].

**Variant discovery and genotyping - SNV and indels**. We processed all case and control BAM files together and performed joint genotyping of SNVs and indels across all samples using GATK (v3.3)[58].

The raw alignments were then processed following GATK best practices with GATK (v3.3). Alignments were realigned around indels using GATK RealignerTargetCreator and IndelRealigner, duplicate marked using Picard MarkDuplicates (v1.120), and base quality scores were recalibrated using GATK BaseRecalibrator. Finally, gVCF files were created for each sample using the GATK HaplotypeCaller (v3.3). Reference files from the GATK v2.8 resource bundle were used throughout. All these steps were coordinated using Piper (v1.4.0).

Joint genotyping was conducted on all cases and controls as recommended by GATK[58]. Due to the large number of samples, 22 batches of 100 samples were merged into 22 separate gVCF files using GATK CombineGVCFs. The 22 individual gVCF files were split by chromosome and further combined with CombineGVCFs. As a result, a single gvcf file was obtained which was used as input for GATK GenotypeGVCF. Subsequently, SNVs and indels were extracted from the resulting gVCF files. To further select high-quality genetic variants, GATK VQSR filtering was executed on SNPs and indels separately using GATK VariantRecalibrator and ApplyRecalibration walkers. VQSR sensitivity thresholds were selected based on maximization of sensitivity of variant discovery in comparison with WES data previously performed on the same samples.

GATK Variant Quality Score Recalibration (VQSR) was used to filter variants as recommended by GATK guidelines. The SNV VQSR model was trained using SNP sites from HapMap3.3[37], 1000 Genomes Project (1000GP) sites found to be polymorphic on Illumina Omni 2.5 M SNP arrays[59], 1000GP Phase 1 high-confidence SNPs[60], and dbSNP[61] (v138). A 99.6% sensitivity threshold was applied to filter variants resulting in a Ti/Tv ratio of 2.001. The indel VQSR model was trained using high-confidence indel sites from[62], 1000GP and dbSNP (v138) and a 99.0% sensitivity threshold was used. The sensitivity thresholds were determined empirically by comparing to WES data in the same samples to optimize sensitivity and specificity of variant detection. We kept only the 'PASS' variants based on results of VQSR.

Variant calling on sex chromosomes was performed separately from the autosomes. GATK Haplotype Caller walker was executed with ploidy = 1 flag on male samples except for PAR regions which were done with ploidy = 2. CombineGVCFs and GenotypeGVCFs were performed by analogy with the processing of the autosomes, see above. VQSR filtering was performed with the sensitivity thresholds inferred from the autosomes. To assess the robustness of the callset, we evaluated hard filters in comparison to VQSR filter. We constructed histograms of 16 variant quality metrics reported by GATK GenotypeGVCFs, manually selected reasonable thresholds for good quality variants, and performed hard filtering according to the selected thresholds. We found that these two filtering strategies, VQSR and hard filtering, gave nearly identical results confirming robustness of the final variant call set.

**Variant discovery and genotyping—structural variants**. We applied three complimentary algorithms for the discovery and genotyping of structural variants (SVs). These algorithms were chosen for their established performance in the 1000GP[32]. We processed all case and control genomes together using protocols recommended by specific algorithms.

We used ExpansionHunter[63] (v2.5.5) with default parameters to identify expansions of short tandem repeats. Using PCR-free WGS, ExpansionHunter can accurately genotype known pathogenic repeat expansions even when the expanded repeat is larger than the read length. With ExpansionHunter v2.5.5, the catalog of known pathogenic repeat expansions covers repeats in 16 genes: *AR*, *ATN1*, *ATXN1*, *ATXN10*, *ATXN2*, *ATXN3*, *ATXN7*, *C9ORF72*, *CACNA1A*, *CSTB*, *DMPK*, *FXN*, *FMR1*, *HTT*, *JPH3*, and *PPP2R2B*. The sizes of the pathogenic repeat expansions are documented in the literature (Table S3). Using the disease thresholds, we identified pathogenic repeat expansions, and the number of cases and the number of controls carrying these pathogenic repeat expansions.

We used Delly[64] (v0.7.7) with default parameters to detect and genotype three types of SV call sets: deletions, tandem duplications, and inversions that are between 500 bp and 500 Mb. We ran the default protocol for germline DNA and high-coverage sequencing. Specifically, for each type of SV, we (1) discover SV sites per sample using paired-end mapping signature and split-read refinement; (2) merge SV sites into a unified site list following strategies used by 1000GP[32] (i.e., for deletions and duplications: 70% reciprocal overlap and a max. breakpoint offset of 250 bp; for inversions: 90% reciprocal overlap and a max. breakpoint offset of 50 bp); (3) genotype the unified SV sites in all samples; (4) merge all genotyped samples to get a single VCF; and (5) apply the default germline SV filters to identify confident SVs (i.e., min. fractional ALT support = 0.2, min. SV size = 500 bp, max. SV size = 500 Mb, min. fraction of genotyped samples = 0.75, min. median GQ for carriers and non-carriers = 15, max. read-depth ratio of carrier vs. non-carrier for a deletion = 0.8, min. read-depth ratio of carrier vs. non-carrier for a duplication = 1.2, and "PASS" variants). Finally, we kept only high-confident genotypes that passed the per-sample genotype filter (i.e., FORMAT/FT = PASS), and had additional support from read-depth-based copy number estimates (i.e., FORMAT/CN < 2 for deletions, CN > 2 for duplications, and CN = 2 for inversion genotypes).

We used the Mobile Element Locator Tool (MELT, v2)[65] to detect and genotype three types of mobile element insertions (MEI) including ALU, SVA, and LINE1. We used the MELT-SPLIT workflow with default parameters which consists four steps: (1) MEI discovery in individual samples; (2) group analysis whereby discovery information are merged across all samples to build models containing all available evidence for each candidate MEI site; (3) genotyping all WGS samples using the merged MEI discovery information; (4) final filtering and merging of individual samples into final VCF. We used the default filters (no-call filter, 5′ and 3′ evidence filter, discordant pair overlap filter, low complexity filter, and allele count 0 filter) and included in the final VCF only those variants that passed the default filtering of MELT.

**Evaluation of variant detection**. For SNV/indels, we used variant calls from exome sequencing to evaluate genotype accuracy from WGS. We focused on the autosomes and estimated genotype accuracy by calculating the concordance rate between WGS-based genotypes and those obtained from exome sequencing across variants that overlapped between the two technologies. We calculated the overall concordance rate as well as concordance rates when WES-based genotypes are homozygous reference, heterozygous, and homozygous non-reference. In all calculations, only genotypes with sequencing depth ≥ 10 and GQ ≥ 20 were included in the comparison. Python code "concordance.py" (https://github.com/jinszatkiewicz/swsczwgs) was used for this analysis.

For deletions and duplications, we evaluated concordance using prior data from GWA SNP array or exome sequencing. Previously GWA genotyping arrays detected large and rare deletions and duplications genome-wide and WES detected rare exonic deletions and duplications in the same samples. We compared the concordance between WGS-based genotypes with those based on either GWAS array or exome sequencing across overlapping variants. Any overlapping variants must have ≥50% reciprocal overlap and occur in the same individual. We calculated the overall concordance rate as well as concordance rates when genotypes from the GWA array or exome sequencing are heterozygous and homozygous non-reference. Variant overlap was performed using BEDTools (v2.28.0).

**Quality control**. For subject quality control, we used PLINK (v1.9). In sum, subject QC excluded 9 subjects for failed sequencing quality metrics (1 case excluded), sex mismatch (1 control excluded), sex chromosomal abnormality (2 cases with XXY excluded), and one of any pair of subjects with high relatedness $\hat{\pi} > 0.2$ (5 controls excluded). These procedures resulted in a final sample size of 2098 subjects (1162 schizophrenia cases and 936 controls), all of whom had SNV/indel missing rate per sample < 0.01 and heterozygosity rate < 0.1. In selection of the schizophrenia cases, we excluded carriers of known large pathogenic CNVs and abnormally high total number of CNVs as identified by Szatkiewicz et al.[18] using SNP arrays. We confirmed this fact using SV calls from WGS.

Sex check was performed using heterozygosity rate of sex chromosomes and by examining the coverage of sex chromosomes. This identified a sex mismatch when the reported sex does not match the biological sex and chromosomal abnormality when extra chromosomes were present. Relatedness testing and principal component analysis (PCA) were done following established pipelines using eligible bi-allelic autosomal SNPs using PLINK (v1.9). Of all bi-allelic autosomal SNPs, we removed variants that had minor allele frequency < 0.05, missing rate per variant >0.01, missing rate per variant in cases and controls >0.02 or $P < 0.005$, Hardy–Weinberg equilibrium false discovery rate (FDR) $< 1\times10^{-6}$ (controls) or $<1\times10^{-10}$ (cases), or were in linkage disequilibrium ($r^2 > 0.05$). Relatedness testing identified any pairs of subjects with $\hat{\pi} > 0.2$, based on which we removed one member of each relative pair. PCA estimated 20 PCs which were used in empirical evaluation of covariates to be included in association analyses. Furthermore, for quality control purpose, we performed PCA of our data together with 1000 Genomes Project data on HapMap individuals and SweGen data on NSPHS individuals. The same quality steps were followed for the identification of eligible SNPs in the combined data.

For SNV/indel quality control, we removed variants if missing rate per variant > 0.01 (before sample removal) and applied genotype QC by setting low quality genotypes with DP < 10 or GQ < 20 as missing. We then removed variants that were: monomorphic, missing rate per variant > 0.02 (after genotype QC and sample removal), missing rate per variant difference in cases and controls >0.02 or $P < 0.005$, Hardy–Weinberg equilibrium FDR $< 1\times10^{-6}$ (controls) or $<1\times10^{-10}$ (cases). After QC, we extracted variants with minor allele frequency (MAF) $\geq 0.01$ for common variant association analysis and the remaining for rare variant aggregated association analysis. These QC procedures were done using PLINK (v1.9).

For SV quality control, we followed established pipelines[8]. SVs were removed if they overlapped by more than 66% with large genome gaps (e.g., centromeres), segmental duplications, or regions subject to somatic V(D)J recombination in white blood cells, with the logic that these variant calls are likely artifactual. Finally, we extracted variants with MAF $\geq 0.01$ for common variant association analysis and the remaining for rare variant aggregated association analysis. These QC procedures were done using PLINK (v1.07).

**Annotation of variants**. We used VEP[66] (v91), vcfanno[67] (v0.2.9), and AnnotSV[68] (v1.1.1) for variant annotations.

For population allele frequency annotations, we annotated SNV/indels using population allele frequencies from gnomAD r2.0.2 genomes and ExAC r0.3 non-psych exomes[24,25]. For SVs, we annotated the variants using population allele frequencies from 1000GP and Database of Genomic Variants (DGV)[31,32]. We used the default settings in AnnotSV, i.e. a SV from 1000GP or DGV is reported if an overlap of >70% is found with a SV to annotate.

For sequence constraints in humans, we annotated variants using the context-dependent tolerance score (CDTS) using the map of sequence constraint for the human species[19]. Files containing CDTS were downloaded from http://www.hli-opendata.com/noncoding/. The downloaded CDTS scores were presented in 10 bp bins in hg38, which was liftover to GRh37/hg19 for the analyses in this study. When a variant spans multiple CDTS bins, mean CDTS was computed and used to annotate the variant. For sequence constraints in mammals, we used the genomic evolutionary rate profiling (GERP) score[20].

For transcript-level annotations, we annotated variants with VEP (v91) using Ensembl transcripts from GENCODE[69] (v16). For SNVs/indels, we further annotated the variants using annotation database dbNSFP 3.5_a. Exonic SNV/indels are classified into groups following criteria based on those used in Genovese et al.[10]: synonymous, missense non-damaging, missense damaging (dbNSFP_MetaSVM_pred = "D" and dbNSFP_fathmm_MKL_coding_pred = "D"), and loss-of-function (stop-gain, frameshift, or splice donor/acceptor).

For brain exons annotations, we obtained a dataset of long-read RNAseq data from a published dataset of long-read RNA sequencing of human brain tissue[29]. The data came in the form of a BED file where each interval represents a uniquely observed exonic region in the data, along with the total number of reads aligning to the region. We took the subset of exons with at least 10 overlapping reads, sufficient support for the exon coming from an isoform that is unlikely to be mere transcriptional noise. We split exons into (1) those within coding loci, and (2) those outside coding loci by simply subsetting intervals on gene-based merged translation start/stop intervals, representing a space where a novel coding exon could potentially be found.

For brain epigenomics annotations, we relied on empirically generated annotations that have shown to be important to gene regulation in the brain.

Epigenomic data are restricted to the autosomes. First, we used the open chromatin regions obtained from ATAC-seq on adult prefrontal cortex brain samples as reported in Bryois et al.[12]. ATAC-seq was performed on adult prefrontal cortex brain samples from 135 individuals with schizophrenia and 137 controls. A total of 118,152 high-confidence ATAC-seq peaks were identified. Second, we used the "easy-HiC" readouts obtained from adult temporal cortex as described in Giusti-Rodríguez et al.[13]. "Easy Hi-C" was applied to six postmortem samples ($N = 3$ adult temporal cortex and $N = 3$ fetal cerebra) and 1.323 billion high-confidence cis-contacts were used for analyses. Three major read-outs were generated including frequently interacting regions (FIREs), chromatin interactions (a.k.a. Hi–C loops), and topologically associating domains (TADs). FIREs were defined as 40-kb genomic bins with significantly more Hi–C interactions (FIRE score $P <$ 0.05). Chromatin interactions were defined as intra-chromosomal chromatin interactions between 10 kb bins that were >20 kb apart (i.e., not contiguous) and ≤2 Mb apart. FIREs are a small subset of all chromatin interactions, which have considerably more three-dimensional contacts. Chromatin interactions have a strong tendency to occur within TADs (discrete megabase-scale regions with less frequent interactions outside the regions). TAD boundaries are defined in 40 kb bins. Finally, we further included epigenetic marks (i.e. CTCF, H3K27ac, and H3K4me3) obtained from ChIP-seq using postmortem brain tissue from fetal and adult samples that were generated in[13].

Using gene model defined by GENCODE (v16), we assessed gene sets previously implicated in schizophrenia and neurodevelopmental disorders including:

- Loss-of-function (LOF) intolerant genes: we used genes from Lek et al.[24].
- Calcium Channel gene set: we used the 26 genes from voltage-dependent calcium channel, available at https://www.genenames.org/cgi-bin/genefamilies/set/253.
- CELF4 gene set: we used genes with "iCLIP occupancy" >0.2 from Supplementary Table 4 of Wagnon et al.[70].
- CHD8 gene set: we used genes from Cotney et al.[71].
- FMRP Darnell gene set: we used the 842 mouse genes from Supplementary Table 2A of Darnell et al.[72], including all genes with FDR < 0.01.
- NMDARC: we used a list of combined NMDAR and ARC complexes genes from Supplementary Table 9 of Kirov et al.[73].
- PSD gene set: we used a gene list generated from human cortex biopsy data from Bayes et al.[74].
- PSD-95 gene set: we used a gene list generated from human cortex biopsy data from Bayes et al.[74].
- RBFOX gene sets: we selected RBFOX1/2/3 genes from Supplementary Table 1 of Weyn-Vanhentenryck et al.[75].
- Genes.ID/DD/ASD: we selected 288 genes implicated in de novo variant studies from Supplementary Tables 15–18 of Nguyen et al.[11], based on $q$-value < 0.05 for developmental delay (DD), $q$-value < 0.1 for autism spectrum disorder (ASD), $q$-value < 0.1 for intellectual disability (ID), and $q$-value < 0.5 for epilepsy (EPI).
- SCZGWAS: genes implicated in schizophrenia common variant association studies, for which we used genes from the 145 regions known to be associated with schizophrenia from Pardinas et al.[6].
- CMCqval05: The CommonMind Consortium (CMC) sequenced RNA from dorsolateral prefrontal cortex of schizophrenia cases ($N = 258$) and control subjects ($N = 279$), from which we selected genes implicated to have differential expression in human brain between cases and controls based on $q$-value < 0.05[76].

For certain tests of SNV/indel burden we focused on burden within gene regions of a generalized coding transcript structure, broadly defined as 35 kb upstream of the most distal transcription start site to 10 kb downstream of the most distal transcription start site (transcript_35kb_10kb).

**Variant subsetting**. Protein coding sequences are defined using protein-coding transcripts from GENCODE (v16). We focused coding SNV/indel analyses on a set of variants which to a high degree of confidence impact bases involved in the production of a functional protein. Coding variants have an at least one transcript-level IMPACT classification of LOW, MODERATE or HIGH according to VEP (v91). We defined noncoding SNV/indels if they did not alter sequence content of coding regions or splice dinucleotides of GENCODE protein-coding transcripts. Noncoding variants only have IMPACT classifications of MODIFIER according to VEP (91).

For SVs, we followed criteria used in Brandler et al.[77] to define coding versus noncoding variants. Protein coding sequences are defined using the consensus coding sequence from GENCODE (v16). Coding deletions, duplications, or mobile element insertions are defined as those affecting any protein-coding sequences. Coding inversions are either having one or both breakpoints inside a protein-coding exon of a gene, or having breakpoints in two different introns of a gene and overlapped with at least one coding exon, or having one breakpoint in an intron of a gene and the other breakpoint outside of that gene. Inversions that inverted an entire gene or genes but had intergenic breakpoints were considered noncoding.

From the post-QC variant callsets, we defined ultra-rare SNV/indels as being a singleton within our WGS cases/control cohort (allele count = 1 in the

2098 post-QC subjects) and absent from independent population cohorts (gnomAD genomes allele count = 0 and non-psychiatric subset of ExAC allele count = 0)[24,25]. This is because the full ExAC and gnomAD exome cohort include exome sequence data derived from schizophrenia case samples included in this study, and applying any MAF constraints using the full cohorts could bias association analysis results against schizophrenia cases. Subsetting of noncoding ultra-rare SNV/Indels on annotations was done using in-house python scripts, VCFscreen v0.1 (https://github.com/Halvee/VCFScreen), based on interval overlap with annotations defined by genomic coordinates.

From the post-QC SV callset, we defined ultra-rare SVs as being single occurrence in our case/control cohort (allele count = 1 in the 2098 post-QC subjects), as well as being absent in independent population cohorts including 1000GP and DGV[31,32]. Based on the default setting of AnnotSV[68], a SV was absent in population cohorts if it did not overlap or overlapped <30% of any variant in the population databases. Subsetting of ultra-rare SVs on annotations was done using PLINK (v1.07) based on interval overlap.

**Power calculation and correction for multiple comparisons**. We used the R/gap package (v1.2.1, https://github.com/jinghuazhao/R) to estimate statistical power for association analyses. We assumed an additive model, lifetime risk of schizophrenia of 1%, and two type I error levels: (1) $5 \times 10^{-8}$ as an established genome-wide significance threshold for single-variant association, (2) $1 \times 10^{-5}$ as in Werling et al.[46]. We computed the minimal detectable genotypic risk ratio to achieve 20%, 80% power over a range of frequency of risk alleles in the population. For single-variant association test, the X-axis of the power plot represents the frequency of a single variant. For burden test, the X-axis of the power plots represents the aggregated frequency of a set of variants aggregated for a target region of interest.

To correct for multiple comparisons in the analysis of common variant association, we used the established genome-wide significance threshold of $5 \times 10^{-8}$. To correct for multiple comparisons in burden analyses of ultra-rare variants, we applied the Benjamini and Hochberg false discovery rate (BH-FDR) method to the family of hypotheses involving ultra-rare SNV/indels which included a total of 74 tests summarized in Supplementary Table 5, and to those involving ultra-rare SVs which included a total of 29 tests summarized in Supplementary Tables 7 and 8. We used the p.adjust function in R (v3.2.2., https://www.r-project.org) to implement the BH-FDR method. We used a threshold of 0.05 on the FDR adjusted P values (a.k.a. q values) to consider statistical significance.

**Burden of ultra-rare SNV/Indels**. Given that the large majority of ultra-rare SNVs and indels (URVs) are not assumed to confer risk in schizophrenia cases, we first tested the null that the total rate of these variants is not a significant predictor of schizophrenia status. Before outlier pruning, with 1162 cases and 936 controls, we fitted a simple logistic regression model with case/control status as the dependent variable and count per sample of URVs as the predictor variable. We found that cases had a higher mean URV count (4456 vs. 4289, P = 0.002, two-sided), and that this was primarily driven by the presence of a portion of samples with unusually high URV counts. The URV outlier samples may be biasing the analysis of URVs even though they were not a concern for the analysis of common variants and SV, and will need to be removed. Following an approach previously established in the full Swedish sample[10], we pruned samples that had an outlier total URV count, here defined as >6000 (Supplementary Fig. 7). The outlier samples appeared to have relatively higher ancestry heterogeneity (Supplementary Fig. 8) similar to the previous finding from the full Swedish sample in Genovese et al.[10]. After outlier pruning, we had 1104 cases and 921 controls and there was no evidence for a difference in mean URV count between cases and controls after this pruning step was carried out (4262 vs 4249, P = 0.4225, one-sided assuming higher burden in cases).

Burden testing was done using VCFscreen (v0.1) and R (v3.2.2). All tests of URV burden in cases relative to controls were carried out using a logistic regression model framework that has been utilized in prior studies[10]. Specifically, the dependent outcome variable in logistic regression is phenotype (schizophrenia = 1, control = 0). The primary predictor is the count per sample of URVs that are specific to target region annotation, whether coding or noncoding. And, based on empirical evaluation, we included three covariate variables in logistic regression: mean_coverage, PC2 (the only PC of the 20 PCs determined from common SNPs which predicted sample case/control status at P < 0.01), and total URV count per sample. We carried out one-sided statistical tests assuming increased burden of URV in cases. Logistic regression models were implemented by the glm function in R (v3.2.2). Odds ratios were computed to measure the increase in the likelihood of having disease per unit increase in URV burden. Empirical P-values were derived by 10,000 permutations by swapping phenotype labels.

**Burden of ultra-rare SVs**. Analysis was done using PLINK (v1.07) and R (v3.2.2). All tests of ultra-rare variant burden in cases relative to controls were carried out using a logistic regression framework that has been established in prior studies[8,18]. Analysis was done for each type of variants separately. In order to ensure the robustness of the analysis, we first empirically evaluated variables that could potential confound association results. We fit a multiple linear regression model, where dependent/outcome variable was the genome-wide total number of ultra-rare SVs, and the independent/predictor variables were sex, mean sequence coverage, and the first three principal components derived from common SNP genotypes. Only the first principal component (PC1) showed significant association with genome-wide burden of ultra-rare SVs. To control its potential confounding effect, we included PC1 as covariate in all tests of burdens of ultra-rare SVs.

For genome-wide burden tests, we fit the following logistic regression model: $y \sim$ covariate + global, where $y$ is the outcome phenotype variable (schizophrenia = 1, control = 0), covariate is the empirically determined covariate variable (i.e. PC1), and global is the genome-wide total number of ultra-rare SVs. For burden tests in target regions, we fit the following logistic regression model for each target region: $y \sim$ covariate + global + target_region, where target_region is the count per sample of ultra-rare SVs that are specific to the target region annotation. Variables global and target_region are computed based on input variants (i.e. coding, noncoding, or combined coding and noncoding). We carried out one-sided statistical tests assuming increased burden of ultra-rare SVs in cases. Logistic regression models were implemented by the glm function in R (v3.2.2). Odds ratios were computed to measure the increase in the likelihood of having disease per unit increase in the burden of ultra-rare SVs. Empirical P values were derived by 10,000 permutations by swapping phenotype labels.

**Single-variant association analysis**. Analysis was done using PLINK (v1.9). Following the general guideline for logistic regression, we used a MAF cutoff of 0.01 to ensure that there were at least 10 events in the less frequent category. Post-QC variants that had MAF > 0.01 were subject for single-variant association analysis. Established variant filters were used to ensure all variants had missing rate per variant < 0.02, missing rate per variant difference in cases and controls < 0.02 (P > 0.005), and Hardy–Weinberg equilibrium FDR < $1 \times 10^{-6}$ (controls) or <$1 \times 10^{-10}$ (cases).

To empirically determine confounding factors, we fit logistic regression models where the dependent outcome variable is phenotype (case/control status) and the independent predictor variables are sex and the first 20 PCs determined from common SNPs. Only PC2 showed significant association with phenotype. Therefore, we included PC2 as covariate for the analysis of autosomal variants, and included PC2 and sex as covariates for the analysis of chromosome X.

A logistic regression model with additive genetic model (Plink –logistic) with empirically determined covariates was used to estimate association between single variants and schizophrenia. Statistical tests were two-sided. The established threshold of $5 \times 10^{-8}$ was used to identify genome-wide significance. Following association, we used IGV to inspect read alignments underlying each putative variant that exceeded the genome-wide significance threshold. False positives that had no IGV support were excluded. Manhattan plots were constructed using R (v3.2.2).

Analysis was done separately for SNV/indels, deletions, duplications, inversions, ALU, SVA, and LINE1. For SNV/indels, deletions, duplications, and inversions, we filtered variants as described above. For ALU, LINE1, and SVA, we additionally restricted our attention to the most reliable variants by selecting variants with a quality score of 5 (best). For ALU, MAF was set to be >0.05 and Hardy–Weinberg equilibrium threshold to FDR < 0.05 for both cases and controls.

**Heritability estimation**. Following Wainschtein et al.[14], we started with the initial set of QC-passing subjects and post-QC SNV/indels and additionally required that each variant observed at least three times in our dataset (i.e. MAF starts at 0.0007). Next we further removed one of each pair of individuals with estimated genetic relatedness >0.05. These procedures resulted in 1151 cases and 911 controls and 17,364,971 sequence variants to be used for narrow-sense heritability estimation. HapMap3 SNPs were downloaded from the HapMap ftp site. To identify imputable variants from Haplotype Reference Consortium (HRC)[21] and 1000 Genomes Project Consortium (1000GP)[22], we used previously imputed data obtained by using SNP genotypes of the schizophrenia subjects from Illumina OmniExpress array genotyping and imputing the array genotype data to the HRC.r1.1 or the 1000GP p3v5 reference panel using EAGLE2[78] (v2.0.5). On the HRC imputation variants, we excluded variants with Imputation Score Info <0.8, individual missing rate >0.05, genotype missing rate >0.05, MAF < 0.0001 and P-value <1e-06 of Hardy–Weinberg equilibrium test. On the 1000GP imputation variants, we excluded variants with Imputation Score Info <0.8, and allele frequencies <0.005 or allele frequencies >0.995 based on previous results[17].

Heritability analysis was done using GCTA[39] (v1.26.0, v1.92.3beta). We assumed lifetime risk of schizophrenia of 1%. We calculated principal components from 1,189,077 HapMap3[37] SNPs selected from the WGS data and included the first 10 PCs (calculated from the same set of HapMap3 SNPs) for the analyses conducted using GCTA's GREML-LDMS[40]. To test the robustness of the estimates, we repeated the analysis while correcting for the first 4 PCs, and for the first 12 PCs and found the results were similar.

With the GREML-LDMS approach, a total of 14 MAF and LD bins were considered and the same set of bins were used for both the imputed-SNPs and for the WGS sequence variants. Specifically, we split the variants into seven different bins based on MAF (0.0007–0.0001, 0.001–0.01, 0.01–0.1, 0.1–0.2, 0.2–0.3, 0.3–0.4, 0.4–0.5) and for each bin of variants, computed SNP-based LD scores with the following parameters: –ld-score-region 200, -ld-wind 10000, ld-rsq-cutoff 0. For a given bin of variants defined by MAF, we defined low LD as < median LD score, and high LD as ≥median LD score. For each bin subsetted on MAF (and further split by LD), we used GCTA to produce a genetic relationship matrix (GRM) from

the set of genotypes. We then used the REML function (via the Fisher scoring algorithm, as implemented in GCTA via –reml-alg 1) to conduct a GREML-LDMS analysis.

**Reporting summary**. Further information on research design is available in the Nature Research Reporting Summary linked to this article.

## Data availability

Summary statistics from single-variant association analysis in this study can be downloaded from Psychiatric Genomics Consortium's website at https://www.med.unc.edu/pgc/download-results/causal-variants-within-scz/. All other summary statistics and supporting data are available in Supplementary Information. Due to recent changes in Swedish and European Union regulations regarding genetic data, we are unable to deposit individual-level data into controlled-access repositories like dbGaP. Collaborative analyses are possible and can be pursued by contacting the authors.

## Code availability

Analysis software used in this study include the following: HiSeq Control Software 3.3.39/RTA 2.7.1; Piper (v1.4.0, https://doi.org/10.5281/zenodo.154586); bwa (v0.7.12, http://bio-bwa.sourceforge.net/); SAMtools (v0.1.19, http://samtools.sourceforge.net/); Picard (v1.120, https://broadinstitute.github.io/picard/); qualimap (v2.2, http://qualimap.bioinfo.cipf.es/); FastQC (v0.11.4, https://www.bioinformatics.babraham.ac.uk/projects/fastqc/); BEDTools (v2.28.0, https://bedtools.readthedocs.io/en/latest/); GATK (v3.3, https://software.broadinstitute.org/gatk/); PLINK (v1.9, https://www.cog-genomics.org/plink/1.9/); PLINK (v1.07, http://zzz.bwh.harvard.edu/plink/); ExpansionHunter (v2.5.5, https://github.com/Illumina/ExpansionHunter); Delly (v0.7.7, https://github.com/dellytools/delly); MELT (v2, http://melt.igs.umaryland.edu/manual.php); VEP (v91, https://github.com/Ensembl/ensembl-vep); vcfanno (v0.2.9, https://github.com/brentp/vcfanno); AnnotSV (v1.1.1, https://lbgi.fr/AnnotSV/); VCFscreen (v0.1, https://github.com/Halvee/VCFScreen); R (v3.2.2, https://www.r-project.org); R/gap package (https://github.com/jinghuazhao/R); GCTA (v1.26.0, v1.92.3beta, https://cnsgenomics.com/software/gcta/#Overview); JMP (v11, https://www.jmp.com/); AbCD Calculator (https://yunliweb.its.unc.edu/abcd_web/AbCD.php). Python code "concordance.py" and other relevant codes are posted at https://github.com/jinszatkiewicz/swsczwgs.

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

## Acknowledgements

This project is funded by NIMH R01 MH106611 (to J.P.S.) and SciLifeLab National Project under the project identifier 2015-R2 (to P.F.S.). P.F.S. gratefully acknowledges support from the Swedish Research Council (Vetenskapsrådet, award D0886501). The Sweden Schizophrenia Study was supported by NIMH R01 MH077139 (to P.F.S.). Prior SNP array and exome sequencing was supported by the Stanley Center of the Broad Institute. Sequencing was performed by the SNP&SEQ Technology Platform in Uppsala. The facility is part of the National Genomics Infrastructure (NGI) Sweden and Science for Life Laboratory. The SNP&SEQ Platform is also supported by the Swedish Research Council and the Knut and Alice Wallenberg Foundation. Sequencing for the SweGen subjects was performed by the National Genomics Infrastructure (NGI). Computational resources from UPPNEX were used for data analysis. Bioinformatics analyses were supported by the Knut and Alice Wallenberg Foundation (2014.0272). The SweGen Project was funded by Science for Life Laboratory (SciLifeLab) as a National Project, supported by the Knut and Alice Wallenberg Foundation (2014.0272), and The National Research Council (PI: U.G.). We acknowledge The Swedish Twin Registry for access to data. The Swedish Twin Registry is managed by Karolinska Institutet and receives funding through the Swedish Research Council under the grant number 2017-00641. Y.L. acknowledges support from NIH U544 HD079124. P.G.R. acknowledges support from NIMH K01 MH109772.

## Author contributions

The study was designed by J.P.S, P.F.S, J.J.C, and Y.L. Funding was obtained by J.P.S. and P.F.S. Statistical analysis was performed by M.H., R.H., J.W., and J.P.S. Bioinformatics analysis was performed by N.O., S.N., P.G.R., and J.P.S. Brain functional genomic annotations were provided by P.G.R. and J.B. Imputation was performed by R.K.; B.N., N.A., and M.F. provided bioinformatics support. A.K.K. interfaced with Swedish national registers and biobanks. A.A., P.K.E.M., U.G., and the SweGen Project provided SweGen data. C.M.H. and P.F.S. provided schizophrenia samples. The manuscript was written by J.P.S., P.F.S., and M.H.; R.H., J.W., N.O., and P.G.R. contributed to writing. All authors reviewed and approved the final version of the manuscript.

## Competing interests

P.F. Sullivan reports the following potentially competing financial interests. Current: Lundbeck (advisory committee, grant recipient). Past three years: Pfizer (scientific advisory board), Element Genomics (consultation fee), and Roche (speaker reimbursement). The other authors declare no competing interests.
