## [Peer Review File · Nature Communications]

Reviewers' Comments:

Reviewer #1:

Remarks to the Author:

The authors have successfully made most of the changes I recommended. Some improvements were argued to be beyond the scope of this manuscript and I concede this point.

Reviewer #2:

Remarks to the Author:

The authors have responded candidly and reasonably to comments from both reviewers. I think the presentation of results is now appropriate.

As both reviewers noted, and the authors agreed, despite the substantial amount of work, but the sample size means it has not revealed much that is new about the genetic architecture of schizophrenia.

On balance I think it's appropriate to publish this updated version so that it can be incorporated effectively into future WGS studies.

NCOMMS-19-41788-T

Response to reviewers' comments:

We thank the reviewers for their positive comments. No changes are requested.

Reviewers' comments:

Reviewer #1 (Remarks to the Author):

The authors have successfully made most of the changes I recommended. Some improvements were argued to be beyond the scope of this manuscript and I concede this point.

Reviewer #2 (Remarks to the Author):

The authors have responded candidly and reasonably to comments from both reviewers. I think the presentation of results is now appropriate.

As both reviewers noted, and the authors agreed, despite the substantial amount of work, but the sample size means it has not revealed much that is new about the genetic architecture of schizophrenia.

On balance I think it's appropriate to publish this updated version so that it can be incorporated effectively into future WGS studies.